# Light-melt adhesive based on dynamic carbon frameworks in a columnar liquid-crystal phase

Shohei Saito[1,2], Shunpei Nobusue[3,†], Eri Tsuzaka[3], Chunxue Yuan[3,†], Chigusa Mori[3], Mitsuo Hara[4], Takahiro Seki[4], Cristopher Camacho[3,5], Stephan Irle[3,6] & Shigehiro Yamaguchi[3,6]

Liquid crystal (LC) provides a suitable platform to exploit structural motions of molecules in a condensed phase. Amplification of the structural changes enables a variety of technologies not only in LC displays but also in other applications. Until very recently, however, a practical use of LCs for removable adhesives has not been explored, although a spontaneous disorganization of LC materials can be easily triggered by light-induced isomerization of photoactive components. The difficulty of such application derives from the requirements for simultaneous implementation of sufficient bonding strength and its rapid disappearance by photoirradiation. Here we report a dynamic molecular LC material that meets these requirements. Columnar-stacked V-shaped carbon frameworks display sufficient bonding strength even during heating conditions, while its bonding ability is immediately lost by a light-induced self-melting function. The light-melt adhesive is reusable and its fluorescence colour reversibly changes during the cycle, visualizing the bonding/nonbonding phases of the adhesive.

[1] Department of Chemistry, Graduate School of Science, Kyoto University, Kyoto 606-8502, Japan. [2] PRESTO, Japan Science and Technology Agency, Kyoto 606-8502, Japan. [3] Department of Chemistry, Graduate School of Science, Nagoya University, Nagoya 464-8602, Japan. [4] Department of Molecular Design and Engineering, Graduate School of Engineering, Nagoya University, Nagoya 464-8603, Japan. [5] School of Chemistry, University of Costa Rica, San Pedro Montes de Oca, San José 11501-2060, Costa Rica. [6] Institute of Transformative Bio-Molecules, Nagoya University, Nagoya 464-8602, Japan. † Present addresses: Department of Materials Engineering Science, Graduate School of Engineering Science, Osaka University, Osaka 560-8531, Japan (S.N.); College of Materials Science and Engineering, Tongji University, Shanghai 201804, China (C.Y.). Correspondence and requests for materials should be addressed to S.S. (email: s_saito@kuchem.kyoto-u.ac.jp).

Liquid crystals (LCs) are versatile condensed materials. Their applications are not limited to LC displays, but covering lasers, photovoltaics, light-emitting diodes (LEDs), field-effect transistors, nonlinear optics, biosensors, switchable windows and nanophotonics[1–3]. Photochemical control of LC materials, as well as polymers and crystals, has long been an attractive subject due to the promising applications[4–8], for example, optical switching and image storage[9–13], photoalignment technique[14–20] and mechanical force induction with photoresponsive actuators[21–30]. In these studies, the unique characteristics of LC materials, the spontaneous amplification of disorganized/ordered phase has been utilized. In some designed systems, ordered packing structures of rod-shaped LC molecules are destroyed by doping a guest molecule with a bent shape[4]. When the guest component is introduced by *in situ* photoisomerization, as often observed in LC azobenzene derivatives, the bulk LC material shows an instant isothermal photoinduced phase transformation, depending on the concentration of the guest dopant. In this context, photoactive LC is a most suitable platform for realizing a quick photomelting (that is, photochemical reaction-triggered isothermal phase transformation into a fluid mixture), while some crystals show a slow photomelting behaviour[31–35]. In spite of this advantage, the application of photomelting LCs to light-removable adhesives has not been explored until very recent reports on azobenzene smectic LCs[36,37], although some polymer materials are developed as photo-deactivatable resins based on various operation principles such as photoinduced crosslinking[38–40], photoacid-catalysed chain modification[41], photochemical cleavage of polymer chains[42] and thermal cleavage of supramolecular chains by light irradiation[43]. The light-melting function offers a new manufacturing technique not realized with conventional hot-melt adhesives[44] as long as the following essential requisites are fulfilled: first, adequate strength for a temporary bond (>1 MPa) even under heating conditions; second, significant reduction of the bonding strengths by light irradiation; and third, quick photoresponse for the separation of bonded materials. It is a formidable challenge to overcome all these essential requisites simultaneously.

Here we report a LC material that satisfies all of the above-mentioned requisites for the light-melt adhesives, namely, a shear strength over 1 MPa up to 110 °C for bonding glass plates, an 85% reduction of the strength by ultraviolet irradiation, and an instant photomelting of the LC film in a few seconds (Supplementary Movie 1). Moreover, this material is reusable as an adhesive, and the transformation between the LC and melted phases is associated with an informative colour change in fluorescence.

## Results

### Molecular design of photoresponsive LC.
Our molecular design of the LC material **1** is based on a unique photoresponsive carbon framework[45,46] with typical dendritic moieties (Fig. 1)[47,48]. The photoresponsive framework is composed of rigid anthracene wings and a flexible joint of cyclooctatetraene[49–51]. This hybrid design of rigidity and flexibility confers two important characteristics on the material. First, the V-shaped molecule with rigid aromatic wings has strong stacking ability to form a columnar array in the condensed phases, which now results in high cohesive force of the LC material for realizing high-temperature resistant bonding. Second, the flexible framework changes its conformation into a flat shape on photoexcitation in the LC phase, which allows the photodimerization of the anthracene moiety[52,53]. This photoresponse leads to the light-induced melting of the LC phase, accompanied by a separation of bonded glass plates.

### Thermal and structural analysis.
The dendritic peripheral chains of **1** caused a LC phase between 65 and 140 °C (Fig. 2a). The LC phase showed only little signs of fluidity, although a shear-induced alignment was confirmed by the polarized optical microscopy (POM; Supplementary Fig. 1). By means of the X-ray diffraction (XRD) analysis, the LC phase was assigned to be a rectangular columnar structure, in which the V-shaped molecules align on top of each other and the stacked arrays are located side by side (Supplementary Figs 2–4). The formation of the columnar structure was supported by the single-crystal X-ray structure analysis of the corresponding derivative **2**, which has no peripheral chains (Fig. 2b; Supplementary Fig. 5). In spite of its non-planarity, the V-shaped framework constructs the twofold π-stacked array with both anthracene wings tightly stacked. The interfacial distance $d(\pi-\pi)$ between the stacked anthracene moieties was observed to be 3.50 Å. The strong intermolecular interaction was also supported by the very large enthalpy change[54] over 30 kJ mol$^{-1}$ in the phase transition of **1** ~140 °C between the columnar LC and isotropic liquid phases (Fig. 2a), which is responsible for the sufficient cohesion strength of this LC material.

### Photoresponse of the LC material.
By the ultraviolet exposure using a 365-nm LED, a LC thin film of **1** between glass plates turned into liquid. The temperature was kept at 100 °C on a hot-stage microscopy. In the POM analysis under crossed Nicols, the bright image of the 5-µm-thick LC film immediately disappeared with 160 mW cm$^{-2}$ irradiance and, instead, a dark field was observed only in the irradiated area, indicative of a photoinduced phase transformation into an isotropic liquid (Fig. 2c). Since this transformation did not take place in the solid phase and significant photochemical reaction was not observed after the photoirradiation at 50 °C (Supplementary Fig. 6), the LC nature is important to induce this photoresponse efficiently. Observation with a thermography camera confirmed a negligible local temperature increase caused by ultraviolet irradiation (Supplementary Fig. 7). The analyses of the resulting liquid revealed that the dimer and trace of oligomers of **1** were produced by light, while unreacted monomer was still present in the melted mixture (Fig. 2d; Supplementary Fig. 8). The photodimer was isolated as a main product and its less symmetric structure was determined based on the $^1$H and $^{13}$C nuclear magnetic resonance analyses (Supplementary Figs 9 and 10). The isolated photodimer showed thermal back reaction, gradually affording the monomer at 130 °C in a solution (Supplementary Fig. 11). Accordingly, on heating the melted mixture at 160 °C for 15 min, the bright POM image of the monomer LC phase was recovered at 100 °C (Supplementary Fig. 12). The recovery of the initial LC structure

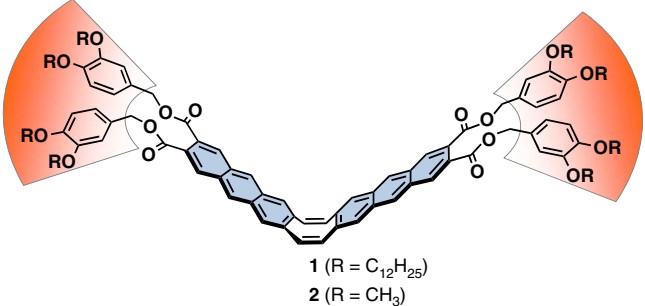

**Figure 1 | Molecular structures of the photoresponsive liquid crystal 1 and its derivative 2.** Rigid anthracene units (blue wings) are fused with a flexible cyclooctatetraene ring (eight-membered ring). Dendritic moieties (orange fans) are attached to the flapping core framework.

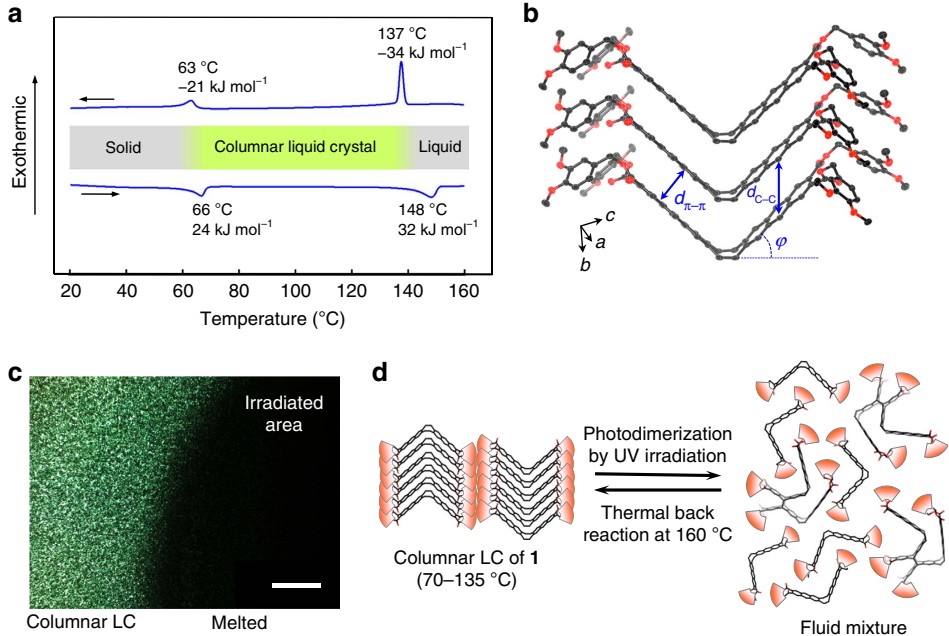

**Figure 2 | Photoinduced melting in a columnar LC phase.** (**a**) Differential scanning calorimetry (DSC) traces of **1** at 2 °C min$^{-1}$ rate of cooling (top) and heating (bottom). (**b**) Crystal packing structure of a derivative **2** with no dendritic peripheral chain. Interfacial distance of the π-stacked anthracene moieties, $d$(π–π) = 3.50 Å. Intermolecular distance between the photoreactive carbon sites, $d$(C–C) = 4.84 Å. Bent angle of the V-shaped molecule, $\varphi$ = 43.8°. (**c**) POM image of the LC film of **1** (bright) and its photoirradiated area (dark) under the crossed Nicols. Scale bar, 500 µm. (**d**) Isothermal photoinduced melting of **1** in the range of 70–135 °C (left to right), in which the columnar LC phase of **1** is photochemically transformed into a fluid mixture mainly composed of unreacted **1** and its photodimer product. Heating the melted mixture at 160 °C induces a thermal back reaction of the photodimer into the monomer **1**, which recovers the columnar LC phase when the temperature is set again in the range of 70–135 °C (right to left).

was confirmed in the grazing incidence XRD analysis of a spin-coated film (Supplementary Fig. 15). Apparently, the photo-induced *in situ* generation of the dimer molecules destabilizes the columnar LC phase of **1** because of their unsuitable shape for the columnar packing[4]. Complete consumption of all monomer units was not necessary for the isothermal photomelting function (Supplementary Fig. 8), presumably because the destruction of the columnar packing is spontaneously induced based on the amplification of the disorganized phase.

**Perfomance of the light-melt adhesive.** Practical performance of **1** as a light-melt adhesive was next demonstrated (Fig. 3). Two non-treated glass plates were bonded by the film of **1**. The glass plates with an attached weight were still bound together over 70 °C by heating with a blow-dryer, at which temperature the phase transition of **1** into the LC phase completed. Once exposed to the ultraviolet LED during the heating, the glass plates were separated in a few seconds and the attached weight dropped (Supplementary Movie 1). However, the ultraviolet exposure at 25 °C did not result in the glass separation even after the 60-s irradiation, which consists with the no significant photoreactive properties in the solid phase of **1**. This inert feature at room temperature is favourable for the avoidance of an unexpected failure under ambient light. Only 2–3 mg of the adhesive was sufficient for bonding two glass plates with 4-cm$^2$ area, from which 8-kg water bottles were suspended (Fig. 3b). Ultimate shear strengths were measured for uniformly prepared 130-µm-thick films (Supplementary Fig. 16). When non-treated glass plates were used, the shear strengths of the film were 1.6 MPa at 25 °C in the solid state, 1.2 MPa at 100 °C, 1.1 MPa at 110 °C and 0.9 MPa at 120 °C in the LC state (Fig. 3c). Notably, the bonding strength still exceeds 1 MPa up to 110 °C, indicating the significant ther-mal resistance. This strength is sufficient for the temporary

bonding of removable adhesives[44]. The shear strength was independent of the hydrophilicity of the glass surface (Fig. 3d). Therefore, the cohesive force, closely related to the intermolecular interaction of **1**, plays a key role in determining the bonding strength of this system rather than the adhesion force, the interaction between the glass surface and the adhesive material (Supplementary Fig. 17)[44].

**Photoinduced melting near the interface.** Despite the strong bonding properties, only after 320 mJ cm$^{-2}$ light exposure at 100 °C the shear strength markedly decreased to 0.2 MPa (Fig. 3c). When the photoirradiation was carried out using a hand-held ultraviolet lamp (365 nm, 3.2 mW cm$^{-2}$), it took ~100 s to induce the photoseparation. On the other hand, quick detachment was achieved in few seconds using ultraviolet LED (365 nm, 160 mW cm$^{-2}$). Since the condensed molecules of **1** efficiently absorb the ultraviolet light, the photoinduced detach-ment took place near the interface between the LC film and the irradiated glass plate. The transmittance exponentially decreased dependent on the thickness of spin-coated films of **1** (Fig. 4a; Supplementary Figs 18 and 19). More than 95% of the 365-nm light was absorbed within 3-µm depth of the film. Consistent with this insight, total irradiation dose required for the glass separation using a 0.5-kg weight was almost constant (320 mJ cm$^{-2}$) regardless of the film thickness over 5 µm (Fig. 4b). From a manufacturing point of view, it is important that the amount of the adhesive residue on the irradiated glass plate is always small even when the thickness of the film is larger (Supplementary Fig. 20).

**Reusability and visualization technology.** The reusability of the adhesive **1** was also demonstrated. After the photoirradiation, the adhesive was heated at 160 °C for 15 min. Then, the shear

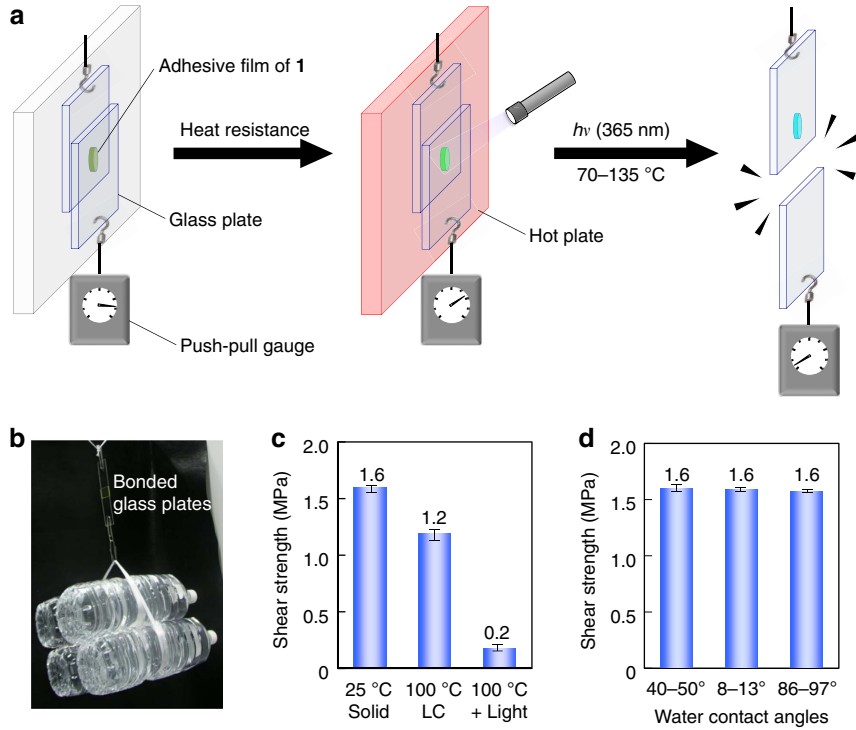

**Figure 3 | Light-melt adhesive properties.** (**a**) High-temperature resistant bonding and photoinduced separation of two glass plates stuck with the adhesive film of **1**. (**b**) Demonstration of the strong adhesive function of **1**. (**c,d**) Ultimate shear strengths of the 130-μm-thick film of **1** depending on the phase of **1** (**c**) and on the hydrophilicity of the glass surface (**d**). See the Methods section for the preparation of the uniform films as well as the hydrophilic and hydrophobic glass surfaces.

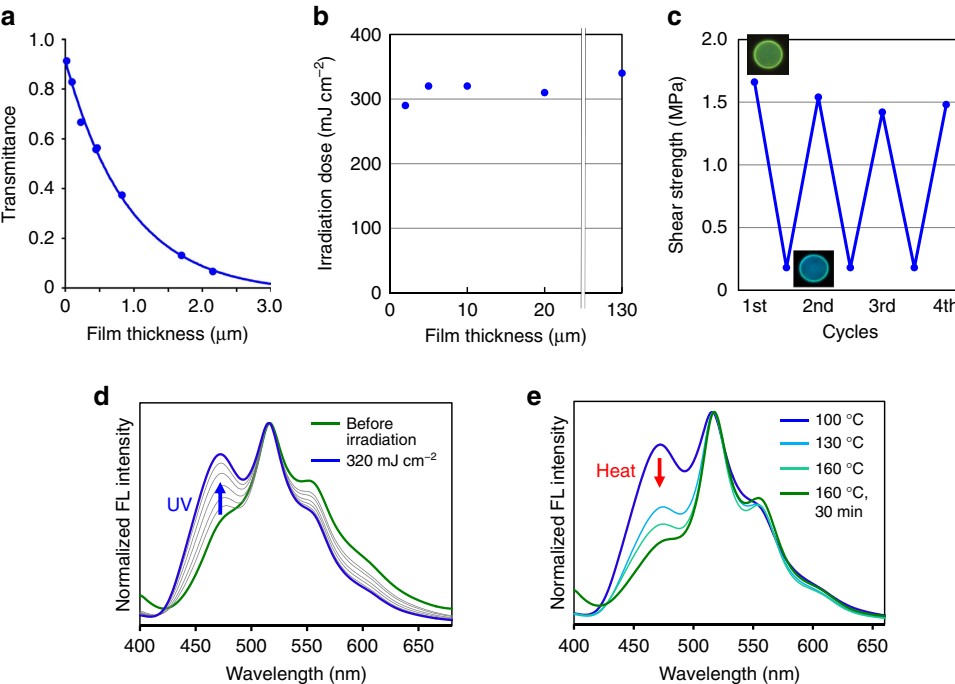

**Figure 4 | Photoresponse and thermal recovery of the film.** (**a**) Transmittance of 365-nm light dependent on the film thickness of **1**. (**b**) Required irradiation dose for glass separation at 100 °C using the films of **1** with different thickness. (**c**) Reusability of the adhesive **1**. The room-temperature shear strengths before and after 320 mJ cm$^{-2}$ ultraviolet exposure at 100 °C in the recycling processes. The inset photographs show the fluorescent film in the corresponding stage. (**d,e**) Fluorescence spectral change during the ultraviolet irradiation on the 5-μm-thick film of **1** (**d**) and its thermal restoring steps (**e**). The film fluorescence at 100 °C before (green line) and after (blue line) light irradiation at 3.2 mW cm$^{-2}$ for 100 s (**d**) and the film fluorescence before (blue line) and after (green line) heating at 160 °C for 30 min (**e**).

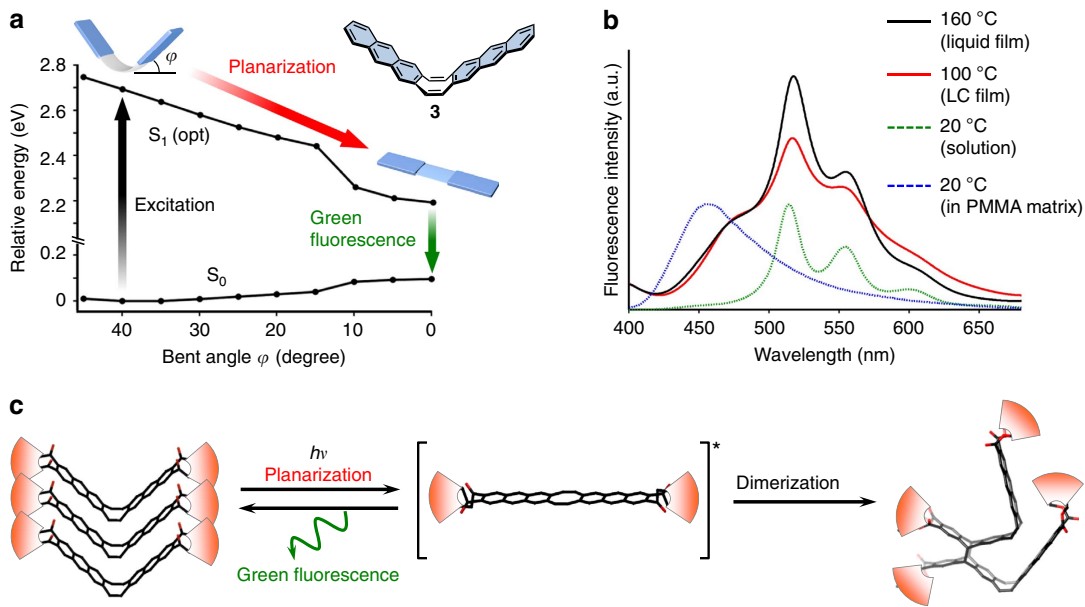

**Figure 5 | Interpretation of the photoresponse mechanism. (a)** Calculated potential energy diagram for the ground state ($S_0$) and lowest excited state ($S_1$) of the carbon framework **3** with fixed bent angle $\varphi$. The relaxed potential energy surface scan was performed for the $S_1$ state of **3** at the TD-PBE0/def-SV(P) level. (**b**) Fluorescence spectra of a thin film of **1** in the liquid phase (160 °C, black line) and in the LC phase (100 °C, red line). Excitation at 365 nm. Fluorescence spectra of **1** in $CH_2Cl_2$ solution (green dotted line) and in polymethyl methacrylate (PMMA) matrix (blue dotted line) are shown for comparison. The fluorescence in PMMA matrix was measured at 20 °C, which is lower than the glass transition temperature of the PMMA ($T_g = 105$ °C). (**c**) The reaction process of the photodimerization of **1** in the LC phase.

strength ~1.5 MPa at 25 °C and the quick photoresponse function at 100 °C were recovered at least four times through this recycling process (Fig. 4c). In addition, the process was accompanied by the fluorescence colour change (Fig. 4d,e), which is useful for the contactless investigation of the adhesive bonding ability. Namely, green fluorescence is a sign of the adhesive recovery in the initializing step, while blue emission of the melted phase informs that the glass plates can be separated. The fluorescence colour change of the LC film observed during the ultraviolet irradiation and the following thermal restoring steps were well explained by the formation of the photodimer of **1** and the recovery of the monomer **1**, respectively (Supplementary Figs 13 and 14).

## Discussion

The mechanism of the photoreaction was studied taking into account the behaviour of **1** in the singlet excited state ($S_1$; Fig. 5). In general, photodimerization of anthracene units proceeds in $S_1$ (ref. 55) only when the reactive sites (the central carbon atoms that form new C–C bonds) approach each other within a distance of 4.2 Å in the condensed phase[53]. By means of the XRD analysis of **1** at 100 °C, the corresponding distance $d$(C–C) in the columnar packing of the V-shaped molecules was estimated to be ca. 4.7 Å, resulting from the slipped alignment of the anthracene moieties (Supplementary Figs 2 and 3), which does not allow the photodimerization. As the significant photoreaction of **1** was not detected at 50 °C (Supplementary Fig. 6), the columnar packing structure is supposed to be fixed in the solid phase (Supplementary Fig. 4). In contrast, the observed photoreactivity in the LC phase indicated that the packing structure was perturbed by the photoexcitation and the dimerization occurred in $S_1$. This interpretation was supported both by theoretical calculations and experimental results. Structural optimization in $S_1$ was performed on the main nanocarbon framework **3**, suggesting that the bent molecule spontaneously changes its conformation in $S_1$ into a planar structure,

because the $S_1$ energy is significantly lowered as the molecular shape becomes flat (Fig. 5a; Supplementary Figs 21 and 22). Fluorescence behaviour of the newly synthesized compound **3** was consistent with the theoretical result. The observed Stokes shift in solution (5,320 cm$^{-1}$) was far larger than those in frozen glass or in polymethyl methacrylate matrix (Supplementary Fig. 23). To elucidate whether the dynamic conformational change of **1** occurs not only in solution but also in the LC phase, the fluorescence spectrum of the LC film was measured. As a result, both the LC film of **1** at 100 °C and the neat liquid of pure compound **1** at 160 °C showed the same green emission band at 515 nm, which was also observed in its solution phase with the large Stokes shift (Fig. 5b; Supplementary Fig. 24). This correspondence indicates that the photoinduced planarization of the molecular framework still took place under these conditions, whereas the dynamic motion was suppressed in polymethyl methacrylate matrix and the V-shaped form of **1** emitted a blue fluorescence band at 460 nm. The blue emission at 475 nm observed in the LC and neat liquid film of **1** can be interpreted as the fluorescence of the shallow V-shaped structure with small bent angle $\varphi$, which was provided because the conformational planarization in $S_1$ was partly suppressed due to the increased viscosity of the LC and neat liquid phases when compared with the low-viscous solution. Overall, the full reaction mechanism of the photomelting process is interpreted as follows (Fig. 5c): near the interface between the LC film of **1** and the glass substrate, the photoexcitation of the stacked V-shaped molecule induces its conformational change into the planar form. As a consequence, the intermolecular distance between the reactive sites of the anthracene moieties is significantly perturbed. Some molecular pairs dimerize photochemically in the $S_1$ excited state, while other excited molecules relax back to the singlet ground state ($S_0$) accompanied by the green fluorescence emitted from the planar form. The photoproducts induce disorganization of the columnar LC phase, leading to the melting behaviour with the loss of bonding strength.

In conclusion, a strong light-melt adhesive based on a photoresponsive columnar LC material has been developed. Tight π-stacking of the V-shaped nanocarbon frameworks resulted in the sufficient bonding properties of the adhesive film even at high temperature. Remarkable photoinduced decrease in the shear strength was realized by the *in situ* dimerization of the LC molecule and the following disorganization of the columnar structure. The fast melting response achieved a quick separation of bonded glass plates leaving the small adhesive residue. We envision that composite materials with the light-melt function will further improve the performance in manufacturing processes, which will accelerate the on-demand photoseparation technology complementary to the other switchable adhesion approaches[56].

## Methods

**Synthesis.** Compounds **1** and **2** were synthesized from a naphthalene dimer[45] having the cyclooctatetraene ring at its centre. Double acene elongation reaction[57] was employed with the corresponding benzyl fumarates. Compound **3** was synthesized through the $[2+2+2+2]$ cycloaddition[58] of a terminal diyne precursor, 2,3-bis(2-propynyl)naphthalene, and the following 2,3-dichloro-5,6-dicyano-1,4-benzoquinone (DDQ) oxidation (Supplementary Methods). The $^1$H and $^{13}$C nuclear magnetic resonance spectra have been displayed for all new compounds (Supplementary Figs 25–30).

**Structural analysis of the LC phase.** XRD measurement of **1** in a Lindemann glass capillary was performed at $100\,^\circ$C using Cu Kα radiation (Supplementary Fig. 2). The LC phase was assigned to be a columnar rectangular phase of $C2/m$ with lattice parameters $a = 68$ Å and $b = 59$ Å (Supplementary Fig. 3). A small peak at $2\theta = 18.86^\circ$, reproducibly observed with the broad signal of the alkyl chain diffraction, was assigned as (001). This signal corresponds to the average interval of the stacking molecules, $d(\text{C–C}) = 4.7$ Å in the LC phase of **1**, which is comparable to that in the crystal packing of **2** (Fig. 2b).

**Single-crystal X-ray structure analysis.** Yellow prism crystals of **2** were obtained by slow diffusion of ethanol into a solution of **2** in anisole. The measurement was performed at 103 K (Supplementary Fig. 5). The structures were solved by a direct method and refined by least-squares calculations on $F^2$ for all independent reflections (SHELXL-2013)[59]. Total 27,088 reflections were collected, among which 5,795 reflections were independent ($R_{int} = 0.0312$). Monoclinic crystal system, space group $C2/c$ (#15), $a = 54.39(5)$, $b = 4.843(4)$, $c = 23.157$ (18) Å, $\beta = 111.690$ (9) °, $V = 5,667$ (8) Å$^3$, $Z = 4$, $T = 103$ K, $R_1 = 0.0367$ ($I > 2\sigma(I)$), $wR_2 = 0.0979$ (all the data), Goodness of fit (GOF) = 1.063.

**Evaluation of the light irradiance.** Ultraviolet irradiation was carried out using a UV-400 series UV-LED (Keyence, UV-50H type, 365 nm) equipped with a UV-L3 lens unit or using a hand-held ultraviolet lamp LUV-16 (AS ONE, 365 nm). Irradiance from these light sources on the sample was measured in advance to be 160 and 3.2 mW cm$^{-2}$, respectively, using an ultraviolet irradiance metre UIT-150 (USHIO).

**Measurement of the ultimate shear strength.** The 130-μm-thick films of **1** were uniformly prepared between glass plates (Supplementary Fig. 16). After the film preparation, one side of the glass plates was connected to a fixed laboratory stand with a strap, while the other side was connected to a digital force gauge. Each measurement was conducted twice and the average values were shown with the error bar. In Fig. 3c, the strength in the LC phase was measured after annealing the sample on the hot plate at $100\,^\circ$C for 15 min. Ultraviolet light irradiation was performed at $100\,^\circ$C for 100 s using a hand-held ultraviolet lamp (365 nm, 3.2 mW cm$^{-2}$), and then the shear strength was measured at the same temperature. In Fig. 2d, non-treated borosilicate glass plates showed a water contact angle of 40–50°. The glass plates with hydrophilic surface (8–13°) were prepared by sonication in a saturated solution of KOH in EtOH for 30 min followed by three times sonication in distilled water for 30 min, while the glass plates with hydrophobic surface (86–97°) were prepared by the exposure of this hydrophilic surface to a HMDS (1,1,1,3,3,3-hexamethyldisilazane) vapour in a sealed tube for > 24 h.

**Measurement of the light transmittance.** To a quartz glass plate ($15 \times 10$ mm) placed on a spin coater, a solution of **1** in CHCl$_3$ with different concentrations (1, 2, 3, 5 and 10 wt%) was dropped so that the solution can cover the entire surface of the glass plate. The sample was rotated on a spin coater at 1,000 r.p.m. for 30 s. The film thickness was determined by the atomic force microscopy analysis to be 14, 96, 220, 460 and 820 nm, respectively (Supplementary Fig. 18). The films with 1,690- and 2,150-nm thicknesses were prepared by overlapping the two and three films coated with the 10 wt% CHCl$_3$ solution of **1**. In consideration of a transmittance loss, the absorbance of the samples at the 365-nm wavelength showed a linear

fitting against the film thickness with the coefficient of determination $R^2 = 0.997$ (Supplementary Fig. 19).

**Theoretical calculations.** Ground- and excited-state density functional theory calculations were performed using the TURBOMOLE program (Fig. 5a; Supplementary Figs 21 and 22): *TURBOMOLE*, ver. 6.3, 2012; a development of University of Karlsruhe and Forschungszentrum Karlsruhe GmbH, 1989 − 2007, TURBOMOLE GmbH, since 2007, available at http://www.turbomole.com.

**Data availability.** The crystallographic data of compound **2** has been deposited in the Cambridge Crystallographic Data Centre with the CCDC number of 1054572. The authors declare that the other data supporting the findings of this study are available on request.

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

## Acknowledgements

This work was supported by PRESTO 'Molecular technology and creation of new functions' from Japan Science and Technology Agency (JST) to S.S., by Grant-in-Aid for Scientific Research on Innovative Areas 'Photosynergetics' (MEXT KAKENHI Grant Number JP15H01083) to S.S. and by Grant-in-Aid for Young Scientists (A) (JSPS KAKENHI Grant Number JP15H05482) to S.S. We thank T. Kato and T. Nishimura (The University of Tokyo) for the fluorescence measurement of the LC thin film. We thank T. Hikage (Nagoya University) for the XRD measurement of the LCs. We thank M. Kotani, S. Suzuki and J. Watanabe (Denka Company Limited) for the valuable advice in the evaluation of shear strength. This work was partly supported by Nanotechnology Platform Program (Molecule and Material Synthesis) of the Ministry of Education, Culture, Sports, Science and Technology (MEXT), Japan.

## Author contributions

S.S. conceived the research and performed the single-crystal X-ray structure analysis: S.N., E.T., and C.M. synthesized the compounds: S.N. evaluated the properties of the adhesive: M.H. and T.S. performed the grazing incidence XRD and atomic force microscopy measurements: C.Y. contributed to the establishment of the synthetic schemes: C.C. and S.I. conducted theoretical calculations and interpreted the results: S.S., S.N. and M.H. co-wrote the paper, and T.S., S.I. and S.Y. proofread it: S.S. designed and directed all the project.

## Additional information

**Competing financial interests:** The authors declare no competing financial interests.

**How to cite this article**: Saito, S. *et al.* Light-melt adhesive based on dynamic carbon frameworks in a columnar liquid-crystal phase. *Nat. Commun.* 7:12094 doi: 10.1038/ncomms12094 (2016).

