## [Peer review file · Nature Communications]

Reviewers' comments:

Reviewer #1 (Remarks to the Author):

The manuscript describes the development of an adhesive that can be removed by heating and illumination with UV light. The material is based on a V-shaped molecule with a cyclooctatetraene core and anthracene wings that exhibit a columnar liquid crystal (LC) phase above room temperature. The adhesive is shown to be stable if illuminated at room temperature, but to lose strength if illuminated in the LC phase. Based on numerous pieces of evidence, the process is attributed to the photo induced dimerization of the molecules, which leads to melting of the LC phase and a decrease in adhesion strength. The paper reports a very interesting use of a photodimerization reaction as a means to affect the phase behavior of a material and at the same time change in a significant manner the adhesion properties of the material. In this implementation, the sample that includes dimers and oligomers exhibits a lower shear strength than the initial monomeric material. The process is reversible and the monomer can be recovered by further heating of the sample.

The paper should be of interest to a broad audience of readers and the concept discussed could lead to advancement in multi-use removable adhesives. The statements and conclusions are generally supported by the wide experimental evidence provided, a good fraction of which is collected in the Supplementary Information. This at times makes it for a difficult read, but it does not seem realistic to include more of the supporting evidence in the main text. Yet, some sentences are confusing and a few of the conclusions not fully supported. These are discussed below and, if revised satisfactorily, the paper may be suitable for publication. Some of the SI information is better in the text and vice versa in any next version (see below).

The conclusions statement "This study well demonstrated the potential of photoresponsive columnar LC materials in the practical use for light-melt adhesives" is too generic. The material discussed here indeed can be switched from a bonding to a non-bonding state when in columnar LC phase, but various other factor seem to have equal importance, including the fact that the photoproduct does not exhibit an LC phase and that it does not revert back to the monomer at the reaction temperature. No evidence in the paper suggests that the light-melt adhesion properties are transferable to other photoresponsive columnar LCs (based on a different molecular mechanism).

It is mentioned that the adhesive properties of compound 1 in the columnar LC phase and the solid phase is ascribed to the tight packing of the V-shaped molecules in stacked arrays. How are these oriented in the films used to test the adhesion properties? Do these depend on the orientation of the molecular stacks relative to the substrates plane?

It is unclear of the difference between the structure of the film at 100C after illumination versus that of the film just heated above 160C. Are they the same or different? The transition of illumination, to heating, to cooling back to 100C needs much further discussion. The authors need to be clear about their use of the term photomelting.

The authors should do proper rheology experiments as a function of temperature with and without illumination.

Some evidence of the role planarization of the molecule in the excited state is discussed on page 12 and seems consistent with the interpretation. But can other possibilities be excluded? For example, could the presence of excimers explain the fluorescence characteristics without involving the conformation changes? And some systems that undergo large conformation changes upon excitation show dual fluorescence, but compound 1 does not: is there evidence from previous work by the authors on other cyclooctatetraene-linked acenes that the conformation changes occur on a timescale competitive with or faster than other relaxation processes (radiative or non-radiative)? If so, it should be cited.

On page 12 it is also mentioned that the photoreaction of 1 did not occur at 60{degree sign}C. Earlier in the paper (page 7) it was only said that no phase transformation took place below 60{degree sign}C. Can evidence be provided to show that no photoproduct was present at 60{degree sign}C? Otherwise it cannot be excluded that the photoreaction took place to some extent, but the presence of dimers was not sufficient to melt the sample (and thus decrease the adhesion). In the latter case the interpretation of the photoreaction mechanism may have to be revised.

The value of the DFT calculations discussed in the paper is rather limited. Only calculations on isolated molecules are considered. It is unlikely that the energetics of the conformation change would be the same in the condensed phase. Indeed, Figure 4b shows that the lowest energy vibronic feature in the fluorescence spectra is around 475 nm for compound 1 in the LC phase or in liquid form, while the one for the compound in solution is around 510 nm. This suggests that either the energy of S1 for the planar conformer is quite different in solution and in neat condensed phases of compound 1, or that other competitive relaxation pathways exist.

What is meant by "thermal initialization" (last but one line on page 10 and various instances in the supplementary material)? Is this the step used for restoring the materials to the monomeric form after irradiation? Or does it refer to the annealing step sometimes used for the samples preparation?

On page 8 it is said that "2-3 mg of materials was sufficient for attaching 8 kg water bottles", but Figure 2b seems to show that the adhesive was used to bond two glass plates from which the bottles are suspended. Was a different experiment performed where the adhesive was directly attached to the bottles?

Unclear statements: "... which consists with the nonreactive properties in the solid phase of 1" (page 8, lines 7-8). "... while the permanent bonding claims higher strengths of 5-50 MPa for structural

adhesive" (page 8, lines 4-5 from the bottom): which permanent bonding is referred to here? is the new material still being discussed or are these target requirements? "We envision that composite materials with light-melt function will further improve the performance in manufacturing processes, which will accelerate the on-demand photoseparation technology complementary to the other switchable adhesion approaches" (last sentence in the conclusions).

In the synthetic details for compounds 1 and 5 (supplementary materials, pages S6-S7), the proton signals in the H-NMR data appear to be over integrated.

In the photoinduced melting of the columnar LC phase of (1), it is not clear if the transition from LC to isotropic liquid is reversible and repeatable. If so, I suggest to characterize the transition dynamics.

The LC optical properties (e.g. long range alignment due to surface functionalization) should be better characterized

Does this LC glue work also for other substrates or is it a specific interaction with this glass surface?

Many times in the text they have mentioned that the LC birefringence was varied, recovered..etc....Did they measure the LC birefringence?

The supposition that 'ordered packing structures of rod-shaped molecules are instantly destroyed... is just not a true statement. Compatibility of a LC phase with any dopant is concentration dependent....

The authors should reference work by Tabiryan/Tabirian on photo-induced isothermal work.

It is unclear why this should be considered a 'smart' liquid crystal.

Reviewer #2 (Remarks to the Author):

Nature Communications manuscript submission NCOMMS-16-00395-T

Title: Light-melt adhesive based on dynamic carbon frameworks in a columnar liquid crystal phase

Authors: Shohei Saito, Shunpei Nobusue, Eri Tsuzaka, Chunxue Yuan, Chigusa Mori, Mitsuo Hara, Takahiro Seki, Cristopher Camacho, Stephan Irle, Shigehiro Yamaguchi

Summary of the key results:

The manuscript describes a reversible adhesive that can be switched between a bonding and a non-bonding state by light. The underlying mechanism is based on a columnar liquid crystal consisting of a V-shaped carbon framework that can be cycled between the ordered, and adhering, state and a disordered, and releasing, state. The material changes from quasi solid to liquid when exposed with light in the columnar phase.

Originality and significance: if not novel, please include reference:

The work is original in the sense that the special use is made of a new liquid crystal molecule that can be switched between V shape in which it forms the ordered state stimulated by pi stacking and a disordered state above a transition around 140 C. The order of the columnar phase can be broken at lower temperature, but still above RT in the columnar phase, by exposure with 365 nm light. This is induced by a change of the conformation of the molecule to more flat shape destroying the columnar organization. To my knowledge the work is original as well as the chosen materials. How significant it is depends on your position, scientific or more practical. The work is scientifically significant as it presents novel switchable molecules (although the photochemistry of anthracene derivatives is known of course). Whether the work is significant for the claimed practical applications is to be seen. The required heating step and subsequent cooling makes that the advantage over e.g. hot melt adhesives is rather limited. It would be different if the authors could switch adhesion at RT. Nevertheless I would like to give the authors the benefit of the doubt. The work may inspire further developments, e.g. to solve the temperature issue. I especially also liked the idea that fluorescence mark the state of the adhesive.

Data & methodology: validity of approach, quality of data, quality of presentation:

The authors support their principle with an extensive and admirable number of analytical data. Already in the main text but also in the Supplementary Information. The mechanism of the photochemistry is well worked out by spectroscopy as well as the data around molecular packing by XRD data. In addition the mechanical studies gave insight in solid-liquid transitions as well as the relevant mechanics for adhesion and release.

Appropriate use of statistics and treatment of uncertainties:

The experiments are not supported by statistics. Nevertheless the results are convincing.

Conclusions: robustness, validity, reliability:

The paper is written in a well, the approach is novel and clearly presented. The results can be verified with the experimental information given by the authors.

Suggested improvements: experiments, data for possible revision:

1. On page 7 the monomer LC phase was recovered at 100 {degree sign}C. In the DSC the transition seems to be reasonable reversible at around 140 C. Please explain.
2. On pages 12 and 13 the authors mention the presence of a polymer matrix which locks the V shape. I find the word polymer somewhat misleading.

References: appropriate credit to previous work:

The manuscript has a complete reference list. Photo-switchable adhesion is more known in biology, e.g. for cell adhesion. The authors may consider to make reference to this.

Clarity and context: lucidity of abstract/summary, appropriateness of abstract, introduction and conclusions:

Introduction and conclusions are clear. The Results and Discussion section is well written. The Supplemental Information is very extensive and supportive, although an explanation of Beer's law goes somewhat far. I recommend publication in Nature Communications.

Reviewer #3 (Remarks to the Author):

This paper deals with photoinduced melting and solidification of LC materials aiming at stimulus-induced adhesive materials. For the light-melt adhesives, some strength is required for a temporary bonding, and quick reduction of the adhesive strength is necessary upon exposure to light. In this paper, the authors achieved enough temporary strength, > 1 MPa, even at 110 °C and quick detachment of the two plates by irradiation. Furthermore, the merit of the present material is that one can see the status of the adhesive material by fluorescence emitted by one of the isomers, which is very useful for practical utilization. All experiments were carried out very carefully and analysis of the experimental results is sound. Basically, I recommend publication of this paper in Nature Communication, but the authors should address to the following issues.

1) p.10, l.3-4: 'The total dose of 320 mJ cm⁻² corresponds to the 2-sec UV-LED irradiation.'

The authors carried out irradiation with a hand-held UV lamp of 3.2 mW cm⁻² light intensity. Then '2-sec irradiation' corresponds to 6.4 mJ cm⁻². What is this discrepancy? For the total dose of 320 mJ cm⁻², they need 100-sec irradiation.

Our reply to the first reviewer's comments:

We are grateful for his positive comments on the broad interests and conceptual advancement of our work. We also thank him for many fruitful suggestions to improve the quality of the manuscript. The point-by-point answers are summarized below.

1. The conclusions statement "This study well demonstrated the potential of photoresponsive columnar LC materials in the practical use for light-melt adhesives" is too generic. The material discussed here indeed can be switched from a bonding to a non-bonding state when in columnar LC phase, but various other factor seem to have equal importance, including the fact that the photoproduct does not exhibit an LC phase and that it does not revert back to the monomer at the reaction temperature. No evidence in the paper suggests that the light-melt adhesion properties are transferable to other photoresponsive columnar LCs (based on a different molecular mechanism).

As the reviewer pointed out, there are some important factors for designing the light-melt adhesives in addition to the photoresponsive properties and the columnar LC structure. Thus we removed this sentence from the conclusion.

*2. It is mentioned that the adhesive properties of compound **1** in the columnar LC phase and the solid phase is ascribed to the tight packing of the V-shaped molecules in stacked arrays. How are*

these oriented in the films used to test the adhesion properties? Do these depend on the orientation of the molecular stacks relative to the substrates plane?

The columnar LC arrays are randomly oriented in the thin film for the shear strength test. The observed bright POM image before the photoirradiation excluded the possibility of a homeotropic alignment (Fig. 2c). No rubbing treatment and no photoalignment technique was employed. In addition, although a specular reflection and a weak diffuse scattering were observed, the 2D XRD pattern is almost angle-independent even for a thinner film (Supplementary Figure 15). We commented about this in the Supplementary Information.

On the other hand, as the reviewer indicated, if the orientation of the columnar arrays would be intentionally induced, the anisotropic bonding strength might be displayed. Those anisotropic adhesive properties are under investigation.

3. It is unclear of the difference between the structure of the film at 100°C after illumination versus that of the film just heated above 160°C. Are they the same or different? The transition of illumination, to heating, to cooling back to 100°C needs much further discussion. The authors need to be clear about their use of the term photomelting.

Although both of the films are optically isotropic fluid, the components of the photoirradiated film are different from that of the film just heated above 160°C. The former is the mixture mainly composed of the unreacted monomer of **1** and its photodimer product (Supplementary Figure 8), whereas the latter is a neat liquid of the pure compound **1**. To distinguish these two phases, the term of “photomelting” was defined in the main text as *an isothermal photoinduced phase transformation into a fluid mixture, which is triggered by the photoreaction of 1*.

[redacted]

4. The authors should do proper rheology experiments as a function of temperature with and without illumination.

We understand the importance of the relationship between adhesive properties and its rheology. In the field of rheology, the adhesion mechanism is actively studied both in molecular to macroscopic scales (for example, Leger, L. & Creton, C. Adhesion mechanisms at soft polymer interfaces, *Phil. Trans. R. Soc. A* **366**, 1425–1442 (2008)). However, the *proper* rheological discussion of our photoresponsive system is not easy at this stage. The film obtained by the photoirradiation has a heterogeneous structure because the light is exponentially absorbed near the interface and the photodimerization proceeds accordingly. The reliable rheological treatment of this photoirradiated film needs another new collaboration with experts in this field, which is to be started.

*5. Some evidence of the role planarization of the molecule in the excited state is discussed on page 12 and seems consistent with the interpretation. But can other possibilities be excluded? For example, could the presence of excimers explain the fluorescence characteristics without involving the conformation changes? And some systems that undergo large conformation changes upon excitation show dual fluorescence, but compound **1** does not: is there evidence from previous work by the authors on other cyclooctatetraene-linked acenes that the*

conformation changes occur on a timescale competitive with or faster than other relaxation processes (radiative or non-radiative)? If so, it should be cited.

The interpretation of the fluorescence spectra was made not only based on the analytical data in this report but also based on our previous work, the references 45 and 46. In the reference 46, the excimer emission of some derivatives has been well investigated in a condensed phase. The fluorescence spectra of the excimer emission were observed as a broad and non-structured band regardless of the type of substituents. This spectral feature of the excimer is different from that of the LC thin film of **1**, which displays the fluorescence band at 516 nm with vibronic structures at 552 nm and 600 nm (shoulder). Importantly, these peak wavelengths (516, 552, 600 nm) clearly correspond to those of the fluorescence bands in solution.

Furthermore, we have an unpublished result of the cyclooctatetraene-linked anthracenes **[redacted]**, which well explains the difference of three emission bands, the blue band (465 nm) from the V-shaped conformer, the green structured bands (516, 557, 601 nm) from the planar conformer, and the more red-shifted broad band from the excimer (640 nm). In a low-viscous solvent **[redacted]**, the conformational change from the V-shaped into the flat structure proceeds in S_1 and then the green emission is observed. With increasing viscosity **[redacted]**, the conformational change is gradually suppressed to show the blue emission (475 nm to 460 nm) of the V-shaped structure. The gradual hypsochromic shift of this blue emission band suggests that the shallow V-shaped structure with small bent angle φ should be considered during the viscosity increase. On the other hand, the broad excimer emission band around 640 nm increases when the molecules get assembled due to the poor solubility **[redacted]**. This spectral evidence strongly supports the interpretation.

To reveal the timescale of the dynamic conformational change, the time-resolved fluorescence spectroscopy and the transient-absorption spectroscopy have been conducted with other collaborators. The unpublished result indicated the time constant of **[redacted]** fluorescence lifetime of the green emission. These results will be reported in another paper.

[redacted]

*6. On page 12 it is also mentioned that the photoreaction of **1** did not occur at 60°C. Earlier in the paper (page 7) it was only said that no phase transformation took place below 60°C. Can evidence be provided to show that no photoproduct was present at 60°C? Otherwise it cannot be excluded that the photoreaction took place to some extent, but the presence of dimers was not sufficient to melt the sample (and thus decrease the adhesion). In the latter case the interpretation of the photoreaction mechanism may have to be revised.*

According to the ^1H NMR analysis **[redacted]**, significant change was not observed after the photoirradiation on the thin film of **1** at 50°C using a hand-held UV lamp (3.2 mW cm^{-2}) for 120 sec. This Figure was added as a Supplementary Figure 6 and the main text was revised accordingly.

[redacted]

7. The value of the DFT calculations discussed in the paper is rather limited. Only calculations only isolated molecules are considered. It is unlikely that the energetics of the conformation change would be the same in the condensed phase. Indeed, Figure 4b shows that the lowest energy vibronic feature in the fluorescence spectra is around 475 nm for compound 1 in the LC phase or in liquid form, while the one for the compound in solution is around 510 nm. This suggests that either the energy of S_1 for the planar conformer is quite different in solution and in neat condensed phases of compound 1, or that other competitive relaxation pathways exist.

We understand that the DFT calculations of the isolated molecule generally do not describe the overall picture of the condensed photoresponsive material. However, it is worth noting here that the main green emission bands at 516, 552, 600 nm observed in the LC and neat liquid phases clearly correspond to those in solution. This agreement indicates that, at least, these green bands can be interpreted without intermolecular interaction.

On the other hand, as the reviewer pointed out, the other blue emission band at around 475 nm was also observed along with the green bands for the LC and neat liquid film of **1**. Importantly, this dual emission feature can be well interpreted in comparison with [redacted] answer to the 5th comment. The fluorescence spectral shapes of the LC and liquid films of **1** are almost identical to those of the diluted solution of the hydrophilic derivative in high-viscous media, [redacted]. Namely, the blue emission at 475 nm can be interpreted as the fluorescence of the shallow V-shaped structure with small bent angle φ , which was provided because the conformational planarization in S_1 was partly suppressed due to the increased viscosity of the LC and neat liquid phases when compared to the low-viscous solution.

Therefore we added the above sentences to the main text in the part of the explanation of the Figure 5b.

8. What is meant by "thermal initialization" (last but one line on page 10 and various instances in the supplementary material)? Is this the step used for restoring the materials to the monomeric form after irradiation? Or does it refer to the annealing step sometimes used for the samples preparation?

The word "thermal initialization" was meant to describe the process for restoring the materials to the monomeric form after irradiation rather than the annealing step for the samples preparation. We replaced the word into *thermal restoring step*.

9. On page 8 it is said that "2-3 mg of materials was sufficient for attaching 8 kg water bottles", but Figure 2b seems to show that the adhesive was used to bond two glass plates from which the bottles are suspended. Was a different experiment performed where the adhesive was directly attached to the bottles?

We revised the corresponding sentence into "Only a 2-3 mg of the adhesive was sufficient for bonding two glass plates with 4 cm² area, from which 8 kg water bottles were suspended (Fig. 3b)".

10. Unclear statements: "... which consists with the nonreactive properties in the solid phase of 1" (page 8, lines 7-8). "... while the permanent bonding claims higher strengths of 5-50 MPa for structural adhesive" (page 8, lines 4-5 from the bottom): which permanent bonding is referred to here? Is the new material still being discussed or are these target requirements? "We envision that composite materials with light-melt function will further improve the performance in manufacturing processes, which will accelerate the on-demand photoseparation technology complementary to the other switchable adhesion approaches" (last sentence in the conclusions).

The classification of removable adhesives is different from structural adhesives in that the former is used for temporary bonding while the latter is used for permanent bonding, for example, the adhesion of some parts in automobiles, aircrafts, and buildings. The bonding strength of 5-50 MPa is normally required for the permanent bonding. However, our study including the future work focuses on the removable adhesives rather than the structural adhesives. Therefore we meant that the bonding strength realized in this light-melt adhesive (more than 1 MPa) is large enough for the purpose of the temporary bonding.

To avoid the misleading statement, we removed the sentence of "... while the permanent bonding claims higher strengths of 5-50 MPa for structural adhesive."

11. In the synthetic details for compounds 1 and 5 (supplementary materials, pages S6-S7), the proton signals in the H-NMR data appear to be overintegrated.

We thank the reviewer for finding the mistake. The integration values of the proton signals were corrected in the page S7 of the Supplementary Information as below.

Synthesis of 1. ...¹H NMR (400 MHz, CDCl₃) δ (ppm) ...1.33–1.23 (m, 128H)...

Synthesis of 5. ...¹H NMR (400 MHz, CDCl₃) δ (ppm) ...6.90 (s, 2H)...

12. In the photoinduced melting of the columnar LC phase of (1), it is not clear if the transition from LC to isotropic liquid is reversible and repeatable. If so, I suggest to characterize the transition dynamics.

The thermal back reaction of the anthracene photodimer into the monomer is known as a clean reaction (Reference 51: Breton, G. W. & Vang, X. Photodimerization of anthracene: a [4π_s + 4π_s] photochemical cycloaddition. *J. Chem. Educ.* **75**, 81–82 (1998)). Indeed, the bright POM image under the crossed Nicols was recovered after heating the photoirradiated isotropic mixture above 160°C for 30 min and the following cooling into the LC temperature range (70–135°C) of **1**. The reversible GI-XRD of the thin film also supported the structural recovery. We added the restored POM image [redacted] in Supplementary Fig. 12.

The transition dynamics is under investigation with other collaborators who have the femtosecond electron diffraction technique as well as the molecular dynamics simulation. The result will be published in another paper.

[redacted]

13. The LC optical properties (e.g. long range alignment due to surface functionalization) should be better characterized.

Quantitative investigation of the polarization and birefringence properties as well as the relationship between the LC alignment and the surface functionalization are in progress.

14. Does this LC glue works also for other substrates or is it a specific interaction with this glass surface?

In this case, the adhesion force at the interface mainly comes from van der Waals interaction between the adhesive and the glass substrate, which was stronger than the cohesive force of the adhesive material regardless of the glass surface hydrophilicity, at least in the range of water contact angle between 10–90° (Fig. 3d and Supplementary Fig. 17). However, the adhesion force normally depends on the materials of the substrates and, more importantly, the removable function is only displayed when the substrate materials are transparent at the excitation wavelength. Overcoming this limitation remains to be done.

15. Many times in the text they have mentioned that the LC birefringence was varied, recovered..etc....Did they measure the LC birefringence?

Since the quantitative birefringence measurement has not been conducted, we replaced the words “LC birefringence” by “*bright POM image under the crossed Nicols.*”

16. The supposition that 'ordered packing structures of rod-shaped molecules are instantly destroyed... ' is just not a true statement. Compatibility of a LC phase with any dopant is concentration dependent....

We understand that the compatibility of a LC phase with any dopant is concentration dependent. To avoid the misleading statement, we revised the sentences as below, in which we mentioned the concentration dependence.

In some designed systems, ordered packing structures of rod-shaped LC molecules are destroyed by doping a guest molecule with a bent shape⁴. When the guest component is introduced by in situ photoisomerization, as often observed in LC azobenzene derivatives, the bulk LC material shows an instant isothermal photoinduced phase transformation, depending on the concentration of the guest dopant.

17. The authors should reference work by Tabiryan/Tabirian on photo-induced isothermal work.

In reference 12, we added the following Prof. Tabiryan’s report on the photoinduced phase transformation.

12. Hrozhyk, U. A., Serak, S. V., Tabiryan, N. V. & Bunning, T. J. Optical tuning of the reflection of cholesterics doped with azobenzene liquid crystals. *Adv. Funct. Mater.* **17**, 1735–1742 (2007).

18. It is unclear why this should be considered a 'smart' liquid crystal.

We removed the word of “smart.”

Our reply to the second reviewer's comments:

We are glad to receive high evaluation on the originality of our molecular system and the idea of the adhesive state visualization by fluorescence. We also thank him for his careful reading of the Supplementary Information (SI). As the reviewer mentioned, the SI also contains many important data. Our answers to his suggested revision are described below.

1. On page 7 the monomer LC phase was recovered at 100 °C. In the DSC the transition seems to be reasonable reversible at around 140 °C. Please explain.

Indeed the bright POM image under crossed Nicols was recovered in the range of the LC phase (70–135°C). We revised the text accordingly.

2. On pages 12 and 13 the authors mention the presence of a polymer matrix which locks the V shape. I find the word polymer somewhat misleading.

We replaced the word “polymer” by “polymethyl methacrylate (PMMA)”. The corresponding sentence was revised as below.

*... the dynamic motion was suppressed in PMMA matrix and the V-shaped form of **1** emitted a blue fluorescence band at 460 nm.*

Fig 5b and its caption were also corrected accordingly, in which we added the comment of the glass transition temperature of the PMMA ($T_g = 105$ °C).

Our reply to the last reviewer's comments:

We appreciate his recommendation of our work for the acceptance. The answer to his comment is attached below.

1) p.10, l.3-4: 'The total dose of 320 mJ cm⁻² corresponds to the 2-sec UV-LED irradiation.'

The authors carried out irradiation with a hand-held UV lamp of 3.2 mW cm⁻² light intensity. Then '2-sec irradiation' corresponds to 6.4 mJ cm⁻². What is this discrepancy? For the total dose of 320 mJ cm⁻², they need 100-sec irradiation.

In this manuscript, we used two types of light sources, a UV-LED (160 mW cm⁻²) and a hand-held UV lamp (3.2 mW cm⁻²). When we use UV-LED (160 mW cm⁻²), it is difficult to accurately determine the total dose required for the glass separation because the irradiation time becomes too short (a few seconds). Therefore we changed the light source into a very weak one, a hand-held UV lamp (3.2 mW cm⁻²), in order to evaluate the required dose. In this case, it almost takes 100 sec to induce the photoseparation, as the reviewer mentioned. (Since this photoseparation event is based on photochemical reaction, the optical linearity must be preserved in this experiment.)

To avoid the misleading statement, we changed the sentences as below.

When the photoirradiation was carried out using a hand-held UV lamp (365 nm, 3.2 mW cm⁻²), it took about 100 sec to induce the photoseparation. On the other hand, quick detachment was achieved in a few seconds by using UV-LED (365 nm, 160 mW cm⁻²).

Other scientific corrections made:

- First uncaptioned figure was labeled as Figure 1 in the revised text. The figure numbers were corrected accordingly.
- Results were divided into the following subheadings, 1) Molecular design of photoresponsive liquid crystal, 2) Thermal and structural analysis, 3) Photoresponse of the LC material, 4) Performance of the light-melt adhesive, 5) Photoinduced melting near the interface, and 6) Reusability and visualization technology.
- References 47 and 48 were added as representative reviews on typical liquid crystalline molecules with dendritic structural design.
- Reference 49 was added as a classical but important review on the dynamic conformational change of the cyclooctatetraene molecules.
- Reference 51 was added for the explanation about the basic chemistry of the anthracene photodimerization.
- In Methods, the crystal system (monoclinic) and the number of the space group *C2/c* (#15) in the international table were added to the section of Single crystal X-ray structure analysis.

Other non-scientific corrections made:

- The affiliation of the corresponding author (Prof. Shohei Saito) was changed since he has moved from Nagoya University to Kyoto University.

- The current address of Dr. Shunpei Nobusue was added.
- In Acknowledgement, DENKA was corrected to Denka Company Limited.

REVIEWERS' COMMENTS:

Reviewer #1 (Remarks to the Author):

The authors have done a sufficient job of addressing this reviewer's comments. The work is publishable and meets the quality and definition needed for Nat. Comm.

Reviewer #2 (Remarks to the Author):

The authors resubmitted their manuscript in answer to reviewers comments. They responded to my remarks adequately although I would have liked to see a rheological study providing information on mechanics and their kinetic aspects. In their rebuttal they addressed this stating that these measurements are within their range of expertise which is a bit awkward for researchers working on adhesive materials. Nevertheless I advice positively towards publication without further changes. It is my opinion that the work is original and might lead to applications for temporary bonding, e.g. in the chips industry. Also they have supported their manuscript extensively with supportive data and references to earlier work on responsive liquid crystals.